# A Mobile Laboratory Robot for Various and Precise Measurements of Crops and Soil in Agricultural Fields: Development and Pilot Study

Shintaro Noda *, Yasunari Miyake, Yuka Nakano, Masayuki Kogoshi, Wataru Iijima and Junichi Nakagawa

Research Center for Agricultural Robotics, National Agriculture and Food Research Organization (NARO), Tsukuba 305-0856, Japan
* Correspondence: nodas152@naro.affrc.go.jp

**Abstract:** Localized management based on multipoint measurements of crops and soil is expected to improve agricultural productivity. The difficulties with this process are as follows: it is time-consuming due to the many measurement points; it requires various sensors for various measurements; it can lead to unstable measurements due to sunlight and wind. To solve the above issues, the system we propose has the advantages of efficient measurements performed by a robot, various measurements performed through exchangeable sensor units, and stable measurements through protecting the crop from sunlight and wind. As a pilot study for the system, we conducted an experiment to simultaneously measure the crops and soil in a cabbage field. The robot achieved mobility in the field, >4 h of operation time, and the ability to obtain soil electrical conductivity measurements and crop imaging at 100 points. Furthermore, the stability of the measurement conditions within the sensor unit during the experiment was evaluated. Compared to the case without the covering, the illuminance became 280-fold stabler (standard deviation = 0.4 lx), and the wind-induced crop shaking became 20-fold lower (root mean square error of the image pixels = 0.5%). The potential impacts of this research include high reproducibility because of the shareable sensor unit and the expectation of new discoveries using precise indoor sensors.

**Keywords:** agricultural robot; fertilization control; EC sensor; light shielding; wind shielding

## 1. Introduction

We are currently conducting research on techniques to improve agricultural productivity through localized maintenance based on multipoint measurements. A report was published in [1] on the results and methods of an experiment of variable-rate fertilization based on the measurement of soil electrical conductivity (EC) in an actual cabbage field. In the experiment, we developed a robot, which we call a mobile laboratory robot, to make the multipoint measurements more efficient. The objective of this paper is to report on its development and evaluate its performance.

Global population growth and climate change have resulted in a strong need for improvements in agricultural productivity [2]. In Japan, the increasing age of workers in the agricultural industry has been accompanied by a striking decline in the number of agricultural workers [3]. Therefore, one approach to improving crop productivity involves the prevention of poor and/or excessive crop growth. More research is needed to develop technologies that can sense variations in crop growth and in the fertilizer content of the soil, as this would help determine the appropriate amount of fertilizer/topdressing required for each crop.

For vegetable crops grown outdoors, fertilization management is generally based on soil diagnostics performed before planting. In the USA, a cultivation method has been developed in which the need for topdressing is determined by measuring the available nitrogen content of soil samples collected from representative points in the field [4]. This





method is widely used in the cultivation of corn, and it has been suggested that this method could also be applied to cabbage [5]. This type of cultivation method employs uniform fertilization based on the analysis of representative soil samples across large fields. However, determining the need for topdressing in smaller parcels of land would allow farmers to respond to uneven fertility within the field, and may help them to reduce variations in crop growth.

The difficulties with multipoint measurements for localized maintenance include the following three points:

1. A large number of measuring points requires a long measurement time;
2. Proper management requires various measurements of the crops and soil;
3. Precise measurements require protection from sunlight and wind effects.

To overcome the above problems, the mobile laboratory robot system we proposed includes the following three features:

1. The robot efficiently performs multipoint measurements to reduce manpower;
2. Exchangeable sensor unit specifications enable various measurements;
3. The crop is protected from sunlight and wind, making measurements precise.

As potential impacts of this research, we expect to improve the reproducibility of agricultural robotics research by specifying the shareable sensor units, and to make new discoveries through the use of precise indoor sensors. An important objective of this paper is to quantify the light/wind shielding performance to determine the availability of precise indoor sensors.

### 1.1. Concept of the Mobile Laboratory Robot

In this paper, we report on a mobile laboratory robot system, which will enable various and precise measurements in the agricultural field (Figure 1).

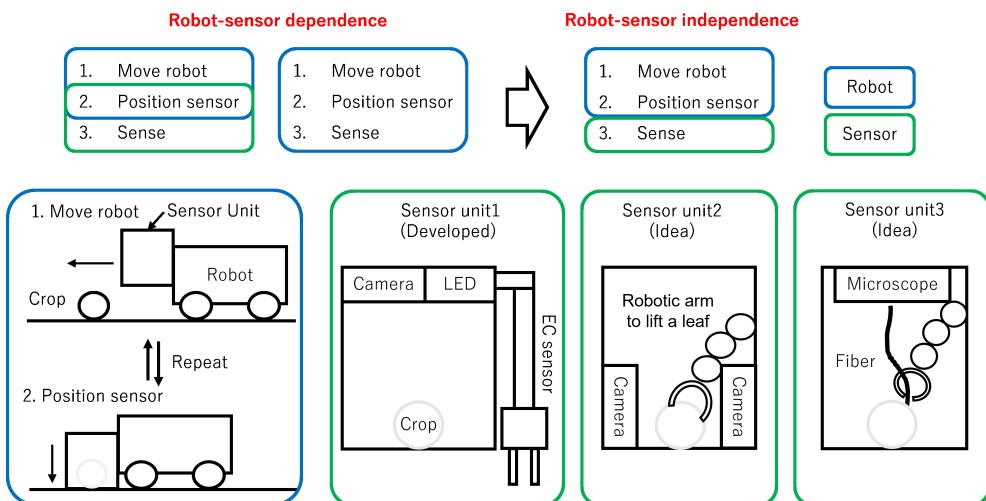

**Figure 1.** The concept of the mobile laboratory robot is to separate the role of the robot (A. moving the robot, and B. positioning the sensor unit) and the role of the sensor unit (C. measuring).

To achieve agricultural measurements using a robot, three processes are required: A. moving the robot; B. positioning the sensor unit; and C. measuring. The mobile laboratory system separates the processes into the role of the robot (A and B) and the role of the sensor unit (C). Furthermore, the sensor unit is box-shaped to cover and shield the entire crop from sunlight and wind. The system has the following two concepts:

1. High extensibility of the sensor unit.
   The system clarifies the specification of the sensor unit, ensuring that the crop is centered and shielded from sunlight and wind. The specification enables the use of

precise indoor sensors that are not intended for outdoor use and facilitate measurements that involve contact with crops using actuators (e.g., small robotic arm);

2.   High reusability of the sensor unit.
      The system separates the role of the robot and the sensor unit. This separation allows the same sensor unit to be used for any robot, and enables the development and evaluation of new sensor units independent of the robot.

As a prototype of the mobile laboratory robot, we developed a four-wheeled robot, as shown in Figure 2, and conducted a pilot study of its performance in an actual agricultural field. The robot was equipped with two RTK-GNSS antennas and a IMU sensor to measure its position and attitude with a cm-level accuracy, and a linear actuator to move the sensor unit to cover the crops.

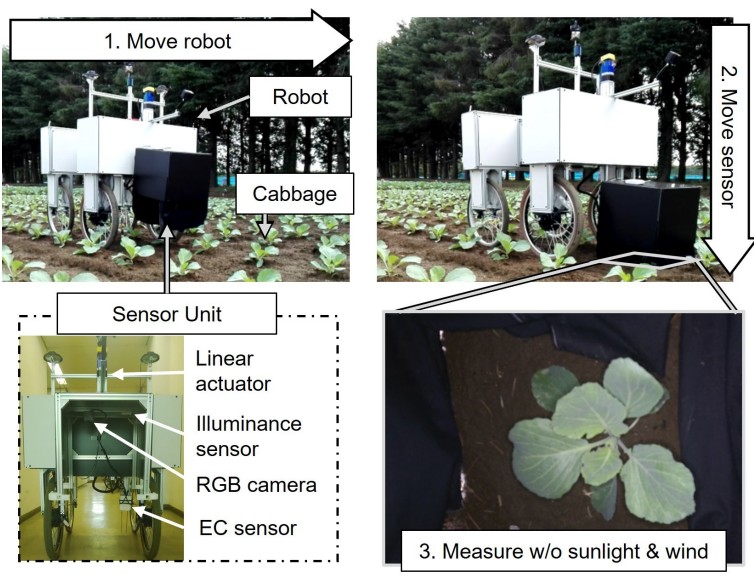

**Figure 2.** Overview of a prototype of the mobile laboratory robot.

As an example of the sensor unit, we developed a box-shaped sensor unit covered with aluminum panels and a light-shielding fabric. The unit included an RGB camera and a soil EC sensor to measure the crops and soil simultaneously. Furthermore, an illuminance sensor was installed alongside the camera and was used to evaluate the light-shielding performance.

The first contribution of this study is the proposal for a mobile laboratory robot system. The system allows precise measurements with indoor sensors outdoors, by positioning the sensor unit and shielding the crop from the surrounding environment. Furthermore, the system allows for the rapid development of modules with various functions by making the sensor unit independent of the robot.

The second practical contribution is the development of a prototype of the mobile laboratory robot and the experimental evaluation in an actual agricultural field. In particular, evaluating the effects of illumination and wind in the sensor unit during the actual measurement process is considered important information for assessing the available sensors. Section 2 summarizes the hardware and software configurations of the robotic system, Section 3 details the findings of our field experiment, and Section 4 provides a summary and discussion.

### 1.2. Related Work

Soil measurements by robots require an actuator that inserts a sensor into the soil. Previous studies performed soil measurements using a linear actuator and penetrometer [6], and a manipulator and a temperature/humidity sensor [7]. However, soil EC is often used

as an indicator of the fertilizer content of soil. Therefore, an EC sensor can also potentially be used for automated measurements.

When conducting measurements in a crop field, it is important to measure the crops themselves (rather than just the soil). Apart from changes in soil nutrient levels, disease-related damage is also an important reason for poor crop growth, and this can be diagnosed using RGB images of crops [8]. Determining whether crop growth is appropriate requires a crop growth model, which can be developed using various factors (such as crop leaf area) [9]. Research on robots with measurement functions, including crop imaging, uses unmanned aerial vehicles [10–12] and unmanned ground vehicles [13–18].

Different objectives require different types of sensing. A top view is needed in order to determine the size of the crop. A side view or a leaf-lifting mechanism is needed to detect the presence of pests on the leaves. Furthermore, dark and windless conditions are important because limited sensors are available for outdoor use. A previous study [18] made imaging conditions more uniform using a shroud to shadow the entire crop. However, there is scope for further research on wind protection and collision avoidance between the shroud and the crop. To enable various and precise measurements, sensors need to be unitized and easily interchangeable, and sunlight and wind in the unit need to be blocked to enable indoor sensors to be used outdoors.

## 2. Materials and Methods

### 2.1. Hardware Configuration

The robotic system is equipped with several sensors and an actuator (Figure 3). The functions of these hardware components are discussed below.

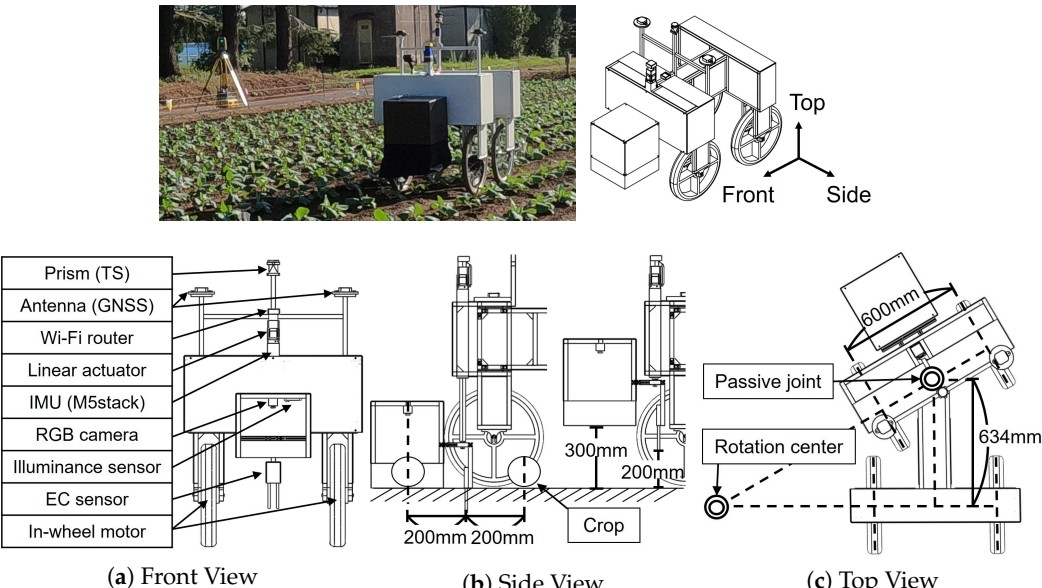

**Figure 3.** Hardware configuration of a mobile laboratory robot. (**a**) Front view to show all hardware components. (**b**) Side view to show the dimensions of the sensor unit. (**c**) Top view to show the dimensions of the steering mechanism.

2.1.1. Ability to Travel between 100 Measurement Points in 4 h without Causing Damage to the Crops in the Field

In the experimental field, cabbage plants were transplanted at 40 cm intervals in ridges placed 60 cm apart (furrow width, 10 cm). The set of dimensions represents a general value in the Japanese cabbage field. Measurements were conducted for up to 3 weeks after transplanting (topdressing stage), when the distance between crop leaves was approximately 10 cm. The field (size, ~1.25 a = 125 m²) contained approximately 1000 crops divided into 20 parcels. During the measurement period, the robotic unit measured soil at 100 points, with 5 points per parcel. The measurement system is discussed in detail below.

The robot travels across the field by straddling the ridge and planted crops. The width of each wheel is 4 cm, which is narrower than the width of the furrow in our experimental field (10 cm). The tread width between the left and right wheels is 60 cm, which was the same as the spaces between the ridges. A passive steering axis is located between the two front wheels (Figure 3c), and the entire system turns based on the difference in speed between the two front wheels. The two front wheels are equipped with in-wheel motors, whereas the two rear wheels are passive. The in-wheel motors are brushless DC motors, and the voltage supplied to these is controlled by a three-phase vector control system using a Hall sensor. The vector control is an open-source implementation called SimpleFOC [19] that runs on an ARM microcontroller.

To facilitate the robot's movement through gaps of several cm, its position must be accurately measured with a 1 cm accuracy. The robot's position and orientation are measured using two RTK-GNSS antennas. Based on the positions of the antennas, the positions of the wheels are calculated using an IMU sensor. A total station with a mounted prism measures the position with a mm-level accuracy and evaluates the RTK-GNSS positioning and traveling accuracy. To allow for >4 h of measuring and traveling, the system is powered by a 213 Wh battery that acts as the power source for the computer and microcontroller. In addition, two 192 Wh batteries power the in-wheel motors. Our field experiment confirmed that this system can operate for just under 6 h.

### 2.1.2. Ability to Carry a Soil EC Sensor and an Actuator That Inserts It into the Soil

The robot was equipped with an EC sensor (Campbell Scientific Inc., Logan, UT, USA, CS655) and a linear actuator (MISUMI Group Inc., Tokyo, Japan, RSD312B) that inserts the EC sensor into the soil. The force required for sensor insertion was measured in advance using a force gauge, and an actuator with a vertical load capacity of >10 kgf was determined to be capable of inserting the sensor into the soil (even dry soil). This linear actuator uses a current limit to push the sensor into the ground with a fixed force, and its range of motion is 30 cm. This allows the robot to avoid damaging the crops while it travels across the field (Figure 3b).

### 2.1.3. Ability to Photograph Crops Using an RGB Camera While Shielding Them from the Effects of Sunlight or Wind

The linear actuator that moves the soil EC sensor also places the sensor unit over the crop and, thus, blocks sunlight and wind. An RGB camera with LED illumination (Shenzhen Ailipu Technology Co., Ltd., Shenzhen, China, ELP-USB4K02AF-KL100W) is installed inside the sensor unit and captures images of the crop. A light-shielding fabric (AS ONE Corp., Osaka, Japan, Light Shielding Fabric (Velvet)) hangs from the bottom of the aluminum box that holds the sensor unit. To account for the unevenness of the ground, this fabric provides additional protection from the sunlight and wind. When the soil EC sensor has been pushed into the ground, there is an extra 5 cm of fabric beyond the ground surface.

When the robot is moving, the sensor unit is raised to a height of 30 cm to prevent collision with the crop (Figure 3b). It is only placed over the crop during measurements. The distance between the soil EC sensor and camera is half the distance between the crops, and this allows the robot to capture the images of the crop and to measure soil EC simultaneously. The light-shielding effect inside the sensor unit is measured with an illuminance sensor (T&D Corp., Nagano, Japan, RTR-574), which is installed beside the camera.

### 2.2. Software Configuration

Figure 4 shows a block diagram of the software system controlling the hardware. Each hardware unit is equipped with individual input/output IO processes that communicate asynchronously and exclusively using a shared memory and semaphores. This system provides the following advantages: (1) improved extensibility and reusability of the software; and (2) real-time control and a low-speed interface.

1. High extensibility and reusability of the software.

   The IO processes are separated into hardware units. Therefore, when a new hardware element is added to the system, the corresponding IO process for that hardware element can simply be added. If the same system is deployed to a new robot, the IO processes corresponding to the common hardware elements can be reused without requiring any changes;

2. High-speed real-time control and low-speed interface.

   High-speed real-time control can be achieved by directly accessing the shared memory on the control computer. At the same time, processes that can be performed at low speeds can be implemented on an external computer via the shared memory and an ROS communication interface ("Bridge" in Figure 4).

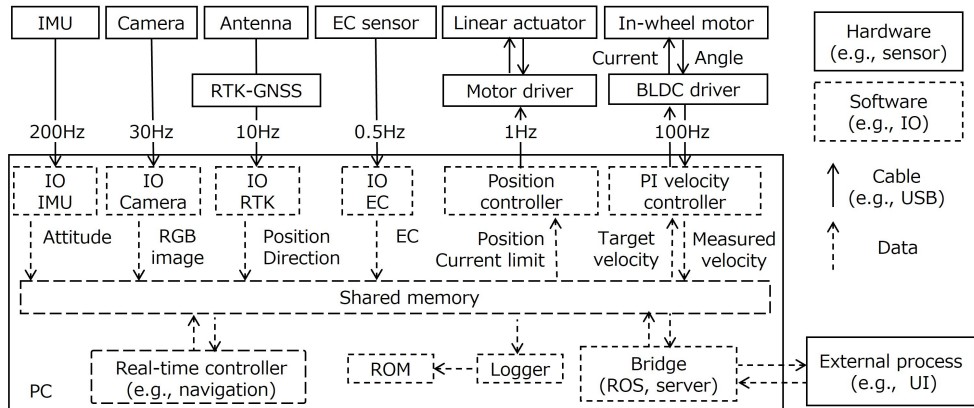

**Figure 4.** Flowchart of the robotic system components.

### 2.3. Navigation System

The robot is controlled using a web server with an ROS interface, as shown in Figure 4. We used open-source software [20,21] for the web server bridging ROS. As shown in Figure 5, the functions of the GUI include displaying a camera image and RTK-GNSS trajectory, controlling the robot's wheels, performing measurements, and specifying the crop ID to store the sensor data. The GUI written in HTML is displayed by accessing the address of the web server PC on the robot from a web browser on an external PC or smartphone. The command of the driving speed was limited by the software to 50 cm/s for safety.

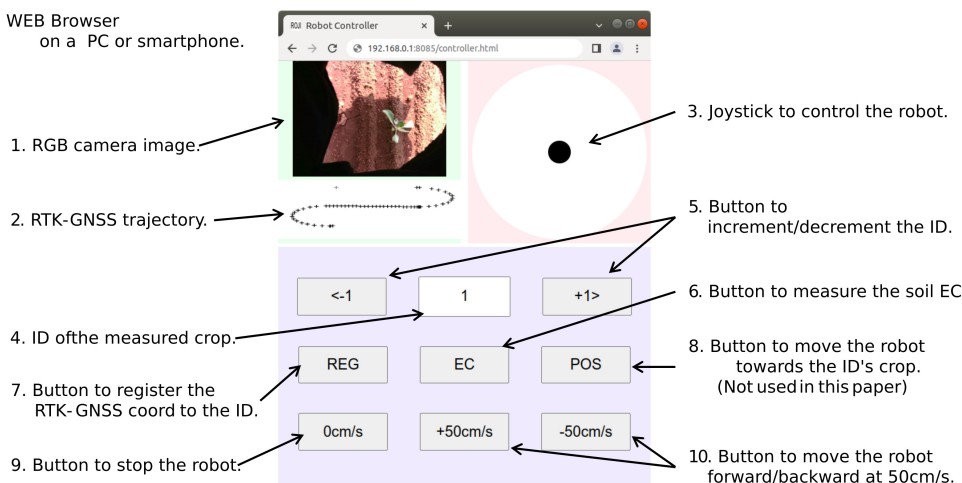

**Figure 5.** Web browser GUI to control the robot, check the sensor data, register the RTK-GNSS coordinates, and measure the soil EC.

Using the browser GUI, the robot is operated according to the following flow, as shown in Figure 6:

1. Check if all crops have been measured, and, if so, terminate the experiment;
2. Increase the ID to identify the crop to be measured next;
3. Move the robot to the position of the crop using the joystick, and check if the crop is positioned in the center of the camera image;
4. Perform the measurements of the soil EC, camera image, and RTK-GNSS coordinate;
5. Go back to 1.

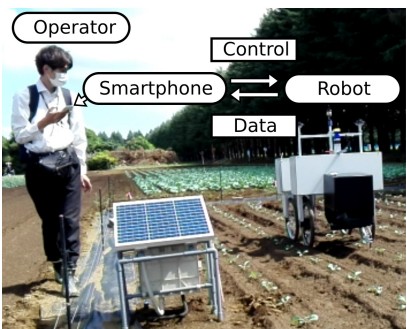
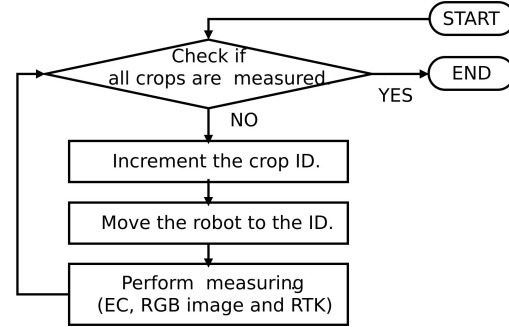

**Figure 6.** Navigation flow to control the robot using the browser GUI.

## 3. Results

### 3.1. Experimental Overview

We evaluated the functioning of the robot in a field experiment conducted at the National Agriculture and Food Research Organization (NARO). During the experiment, the robot was used to perform the following functions simultaneously: measure the soil EC, capture images of the crops, and measure the illuminance inside the sensor unit. Figure 7 shows an aerial photograph of the experimental field. The robot traveled between the 100 measurement points (indicated by crosses), performing 1 min measurements at each point. The following experimental data were taken on 9 December 2022. Similar experiments were conducted on 14, 22, 26, and 30.

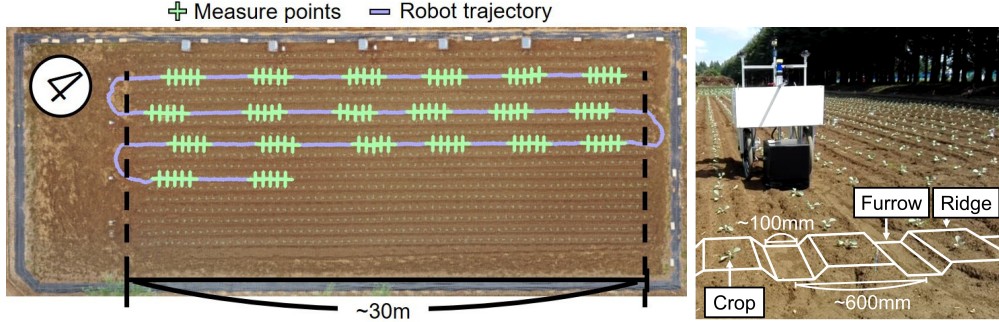

**Figure 7.** Aerial view of the experimental field (**left**) and view of the experiment in progress (**right**).

The raw data are shown in Appendix A. Tables A1 and A2 show all of the measured values. The robot successfully completed a 4 h experiment to capture images of 100 plants, measure the surrounding soil, and obtain the RTK-GNSS coordinates of the robot. The soil measurements were taken every 2 s for 1 min. We used the median value of the last 10 data points to wait for the sensor value to become stable as the sensor body adjusted to the ground temperature.

The projected leaf area was calculated by counting the number of pixels with an a-channel value <130/255 in the LAB color space of the captured image (Figure A1), which was converted to area based on the camera height and the angle of the view. The latitude and longitude were measured at 10 Hz to calculate the average value.

Time-lapse photographs of the robot during the experiment are shown in Figure A2. The weather was clear, although clouds sometimes covered the sun (e.g., ID∼80). The robot was moved by a human joystick control using the web browser GUI.

### 3.2. Light-Shielding Effect of the Sensor Unit

Images of an individual cabbage were captured with the sensor unit lowered onto the plant (Figure 8, left) and raised to a height of 30 cm above the ground (Figure 8, right). When the unit was raised, the robot's body and wheels cast shadows on the plant, and the color of the plant varied by location. In images captured with the sensor unit covering the crop, the lighting conditions were improved because the sensor unit blocked the sunlight.

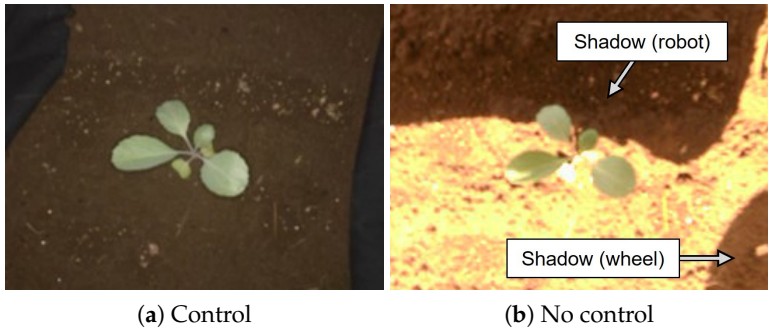

(**a**) Control                        (**b**) No control

**Figure 8.** Images of cabbage plants, as captured by the robot. (**a**) Image captured with the sensor unit covering the crop. (**b**) Image captured with the sensor unit raised to a height of 30 cm above the ground.

The measurements recorded by the illuminance sensor were analyzed to quantify the lighting conditions (Figure 9). When the sensor unit was raised, the illuminance varied greatly (ranging from several tens of lux to >400 lux; Figure 9, dashed line). These fluctuations can be attributed to the effects of clouds and other factors dependent on the direction of travel of the robot. On the day of the experiment, the sky was clear with occasional clouds. Therefore, there were instances during the experiment when the sun was covered by clouds, resulting in lower illuminance. Additionally, the relative positions of the sensor and the robot changed every time the robot changed its direction of travel. This may also have caused differences in shadow formation, leading to changes in illuminance levels.

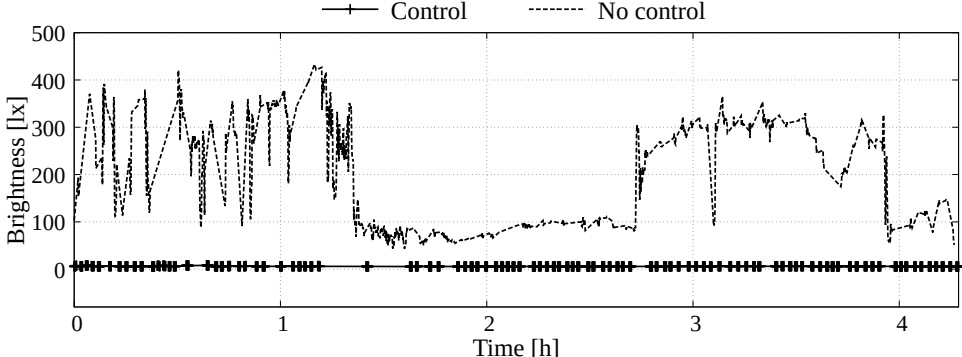

**Figure 9.** Measurements recorded by the illuminance sensor over a period of >4 h, during which the robot traveled across the experimental field and conducted soil measurements at 100 points.

Figure 10 shows the histograms of illuminance levels. When the sensor unit was raised (left; mean illuminance, 196.7 lx; standard deviation, 112.2 lx; minimum, 43.0 lx; maximum, 431.8 lx), the illuminance levels showed large fluctuations. The two peaks (Figure 10b) were considered to correspond to the presence or absence of shadows (caused by clouds or the position of the robot). When the sensor unit covered the crop (mean illuminance, 6.0 lx; standard deviation, 0.4 lx; minimum, 5.5 lx; maximum, 7.9 lx), the illuminance levels were

relatively stable. A comparison of the standard deviations revealed that using the sensor unit could improve the stability of illumination levels 280-fold.

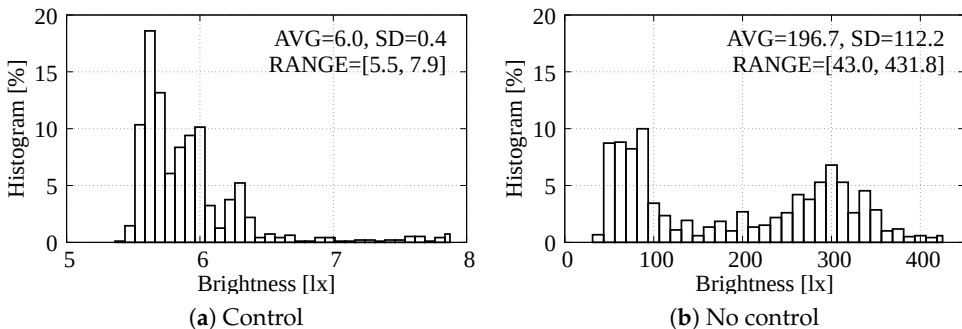

**Figure 10.** Histograms of values recorded by the illuminance sensor. (**a**) Values recorded with the sensor unit covering the crop (mean illuminance, 6.0 lx; standard deviation, 0.4 lx). (**b**) Values measured with the senor unit raised 30 cm above the ground, showing large fluctuations (mean illuminance, 196.7 lx; standard deviation, 112.2 lx).

### 3.3. Wind-Shielding Effect of the Sensor Unit

To confirm the wind-shielding effect of the sensor unit, we estimated the maximum distance between consecutive images obtained from the video of an individual cabbage plant (captured at 10 fps). The wind speed at a 2 m height from the ground surface was 1.1–2.4 m/s between 10:00 and 14:00 [22]. The following equation was used to measure the root mean square error (RMSE) as an indicator of the distance between two images:

$$RMSE(I^1, I^2) = \sqrt{\sum_{c=1}^{3} \sum_{w=1}^{W} \sum_{h=1}^{H} \frac{(I^1_{w,h,c} - I^2_{w,h,c})^2}{3WH}} \tag{1}$$

The two images ($I^1$, $I^2$) were arrays with a total of $3WH$ elements, comprising height ($H$), width ($W$), and RGB values. We scaled the RGB values so that each value was below 100, but not below 0. Based on these values, the RMSE was converted to % units (Figure 11). Two images were captured with the sensor unit covering the plant, and the RMSE between these images was 0.5% (Figure 11a). However, when the sensor unit was not used, the RMSE between the two images was more than 20-fold greater (10.8%). These findings confirmed that the wind caused a considerable movement in the leaves of cabbage plants and that the sensor unit prevented this movement.

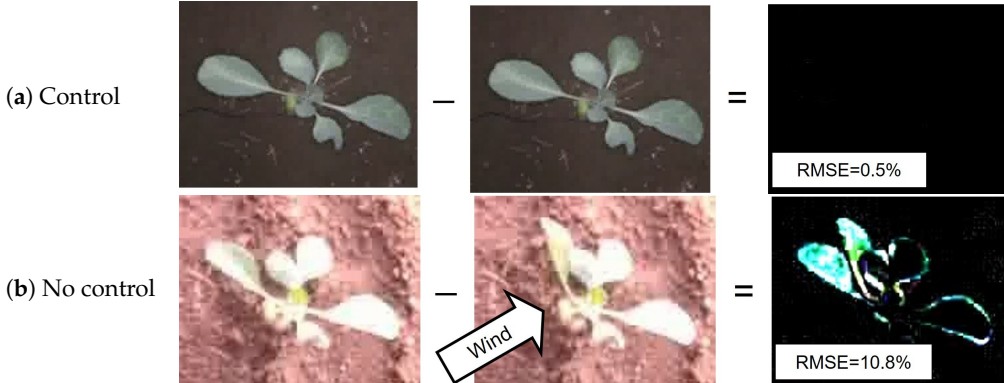

**Figure 11.** Images of an individual cabbage plant with a maximum difference of 0.1 s between consecutive images, and the subtracted image. (**a**) With the sensor unit covering the plant, the root mean of the squared error (RMSE) between the images was 0.5%. (**b**) With the sensor unit not being used, the leaves are shaken by the wind and the maximum RMSE was 10.8% (20-fold greater).

### 3.4. Soil EC Sensing

We visually confirmed that the soil EC data had been recorded at a total of 100 measurement points and recorded correctly (with the sensor inserted into the ground down to the root). The current limit of the linear actuator was set to a value equivalent to 5 kgf.

Figure 12 shows the relationship between the EC values of the soil solution (calculated from the EC values measured by the sensor) and the fertilizer N rate. Each parcel of land in the experimental field had been fertilized with different amounts of a basal fertilizer. Accordingly, the soil in the different parcels had four different N amounts per 10 a (=1000 m²): 0 kg, 9 kg, 12 kg, and 15 kg. The greater the amount of basal fertilizer, the higher the EC value measured, suggesting that the amount of fertilizer applied can be determined based on the EC value measured by the sensor. We conducted the same experiment five times on different days to control the amount of fertilization based on the robotic measurements [1].

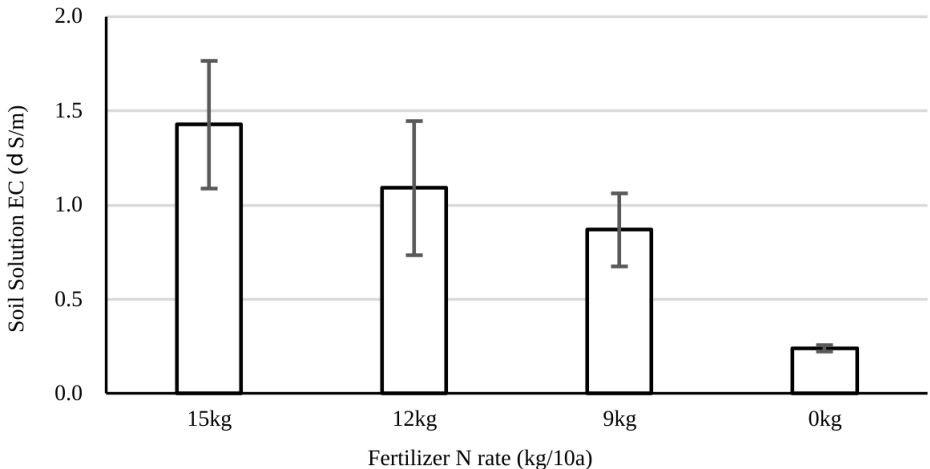

**Figure 12.** Relationship between the electrical conductivity (EC) in the soil solution (vertical axis) and the basal fertilizer rate (horizontal axis). The error bars represent the standard deviation.

The EC of the soil solution ($G_a$)—indicating the expected EC of homogeneous soil—can be calculated based on the EC ($G$) measured in heterogeneous soil. We calculated the $G_a$ using the following equation based on the Hilhorst method [23]:

$$G_a = G \times \frac{\epsilon_w}{\epsilon - \epsilon_d} \tag{2}$$

where $\epsilon$ is the permittivity of the soil (as measured by the EC sensor), $\epsilon_d$ is the permittivity (equal to 2.825, measured after drying the soil), and $\epsilon_w$ is the permittivity of pure water (a fixed value with temperature characteristics). Using the soil temperature ($t$) measured by the EC sensor, we calculated $\epsilon_w$ using the following regression equation [24]:

$$\epsilon_w = 88.15 - 0.414\,t + 0.131 \times 10^{-2}\,t^2 - 0.046 \times 10^{-4}\,t^3 \tag{3}$$

### 3.5. Throughput of the Measurements and Movements of the Robot

As shown in Figure 13, the system repeated the following four procedures:

1.  Move the robot until the crop is centered in the camera;
2.  Move the sensor unit down to cover the crop;
3.  Measure the soil EC, camera image, and RTK-GNSS coordinate;
4.  Move the sensor unit up.

The throughput of each procedure was as follows: 1. 80 s, 2. 13 s, 3. 60 s, and 4. 2 s. Procedures 2 and 4 were paired operations; however, the former required extra time to slowly push the sensor unit to the ground until the load applied to the linear actuator becomes stable (5 kgf). In procedure 3, the robot and linear actuator were stopped

to measure the soil EC, crop image, and RTK-GNSS coordinate. Procedures 2–4 were automatically performed in succession after the operator pressed the EC button on the WEB GUI.

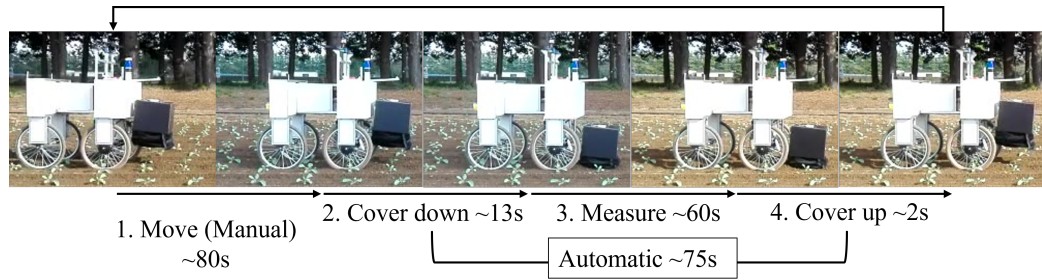

**Figure 13.** Throughput of all experimental procedures including the movements of the robot, moving the the sensor unit down to cover, measuring, and moving the sensor unit up.

Procedure 1 was performed via human control in this experiment. The total travel distance of the robot was ~100 m for 100 cabbages (~1 m for 1 cabbage) and the driving speed of the robot was limited by the software to 50 cm/s. Therefore, the minimum throughput of procedure 1 was 2 s. The reasons for the remaining 78 s include the following: the operator repeatedly adjusted the position of the robot so that the crop was in the center of the camera; and the operator frequently checked if the software was working properly (e.g., data logging).

## 4. Discussion

As shown in Section 3, we confirmed that the mobile laboratory robot is capable of moving around in an actual agricultural field, sensing for 4 h, photographing 100 plants, measuring the surrounding soil, and measuring the position of the plants. Because the remaining battery power at the end of the experiment was about 20%, a ~5 h experiment is possible. Assuming that a prototype of the mobile laboratory system has been completed, our future plans include the use of other sensor units (e.g., hyperspectral camera and robotic arm) and the verification of driving on steeper and uneven terrain (note: the field that the robot traveled in was in good condition without ground tilt and human footprints).

As shown in Figure 10, the illuminance inside the sensor unit could be controlled to $3\sigma = 1.2$ lx. Since typical indoor illuminance ranges from several hundred lux to a thousand lux [25], the effect of sunlight on illuminance inside the sensor unit can be controlled to an error of less than 1% (when using only an ordinary indoor light as a light source). If a light source with high illuminance (exceeding 1000 lux) is used, an error of 0.1% is possible. This indicates that our system allows for high-precision light sensing.

Because this system provides a highly accurate control of illuminance, it can used for several purposes, including imaging using a spectrophotometer or hyperspectral camera in outdoor conditions. In a previous study [26], the researchers buried spectrophotometers mounted on an agricultural machine in the soil and pulled them along for measurements. Using this system, the researchers measured light reflectance (ranging from the near-infrared to UV wavelengths) with a high resolution, while maintaining illuminance. Based on the measurements, they created a regression model that correlated with the nitric acid and phosphorus contents of the soil. The same could be performed for crops on the soil surface using the sensor unit. Its potential applications include the supplementation of nutrient deficiencies in crops via topdressing.

Because the sensor unit provides protection from the wind, it makes it easier to perform specific operations on plants using sprayers or robotic arms. The robot-based droplet application of agricultural chemicals reduces the amount of chemicals used to 1/10 of the conventional amount [27]. However, wind has significant impacts on droplet application. Therefore, the creation of a windless condition inside the sensor unit may allow for more accurate and efficient applications of agricultural chemicals to plants. Additionally, some

sensors enclose the leaf during measurements (e.g., Konica Minolta, Inc., Tokyo, Japan, SPAD-502Plus). A lack of wind-related leaf movement would make it easier to record measurements using these sensors.

In our experiment, the robot was operated by a human. The complete automation of measurements would require the use of high-precision automatic driving technology. In the cabbage field used in this experiment, the furrow width was ~10 cm and the wheel width was 4 cm. In this case, the horizontal alignment error had to be <3 cm, otherwise the robot would have collided with the ridge. Three weeks after transplantation, the gap between the plants was still ~10 cm. Therefore, the positioning accuracy for inserting the sensor into the soil would also need to be in the same order of magnitude (several cm). The studies that use RTK-GNSS antennas for automatic driving in fields [28] report an average error of ~5 cm and a maximum error of ~10 cm. As such, there is scope for further research into developing more precise driving technologies. We have already begun research on high-precision automatic driving and are preparing to publish our results.

By automating the driving, the measurement throughput is expected to become twice as fast. As shown in Section 3.5, the throughput of the automated measurement part is ~75 s and driving by human operation is ~80 s. However, the 80 s includes reducible time to check the software log debugging and to adjust the robot's position. The time will become unnecessary as a result of a stable and high-precision automated driving system. The throughput of the driving part has room for speeding up to 2 s because the average driving distance is ~1 m, and the maximum driving speed is limited by software to 50 cm/s. In that case, the time required for the experiment becomes $(2 + 75) \times 100 = 7700$ s (~2.1 h).

In this study, the robot recorded various types of data. However, performing analyses and controlling the robot using RGB images requires further research, and the development of a technology that recognizes distinct regions within crop images. Maintaining constant lighting conditions (as in the sensor unit) makes it possible to extract the green region of crops using a simple color filter that applies a threshold on the 'a' channel (e.g., in LAB color space). However, more accurate analyses would require collaborations with relevant research teams and the use of deep learning technology [29]. These developments would make it possible to detect withered or discolored leaves, and differentiate between weeds and crops based on the images captured by the robot.

In this study, the distance between wheels was fixed at 60 cm, which is a commonly used value for the distance between ridges in Japanese cabbage fields. However, a mechanism to change the distance between the wheels is needed in order to operate in fields with more general dimensions. Previous studies have shown a method using modular assembly robots [17] to easily reconfigure the distance between wheel modules, and a rotating mechanism to open and close the supporting legs of the wheels [18].

The assumed crops that our system can be adapted to include cabbage, lettuce, and potato. To apply to taller crops (e.g., okra, eggplant, tomato, and cucumber), the sensor unit needs to be extended vertically. Furthermore, holes may need to be drilled to allow air to escape from the inside to prevent the crop from collapsing. Our system is not applicable to crops with a narrow furrow for the robot to enter (e.g., wheat and green onion). However, it may be possible to combine a crane-type agricultural automation system (e.g., FarmBot [30]) with sensor units.

## 5. Conclusions

Here, we report a robotic system that automatically measures the fertilizer content of a field and surveys crop growth. This system can greatly improve agricultural productivity by controlling the amount of fertilizer applied to soil. The functioning and measuring capabilities of this system were evaluated in a field experiment, where the robot accurately measured the light- and wind-shielding effects of the sensor system, photographed the plants, and measured the soil EC.

The system is a mobile laboratory robot system, which enables various and precise measurements in agricultural fields. The system clarifies the sensor unit specifications, meaning that the crop is centrally located and shielded from sunlight and wind; thus, it becomes possible to use precise indoor sensors outdoors, and to facilitate measurements involving contact with crops using a robotic arm. Furthermore, the system clarifies that the positioning of the sensor unit is the role of the robot; thus, it becomes possible to accelerate the development of the sensor unit as a standalone unit independent from the robot.

The software for this system consists of shared memory and independent IO processes that simplify the addition of new sensors. The software configuration implements both high-speed real-time control and a low-speed ROS interface. The hardware includes two RTK-GNSS antennas, a soil EC sensor, an RGB camera, and a linear actuator that moves the sensor up and down. These features allow the robot to travel across the field with a cm-level accuracy while measuring the soil EC and capturing images of the crops.

As per the experimental results, the robot successfully ran for over 4 h, photographed 100 plants, and measured the EC of the surrounding soil at the same time. We visually confirmed that the sensor was inserted correctly into the soil. As per the results of the EC sensing, we showed that the lower the fertilizer N rate of the soil was, the lower the EC value was. We repeated the same experiment five times on different days to control the amount of fertilizer applied [1].

Furthermore, we evaluated the shielding effects of the sensor unit in the field experiment. The results showed that using the sensor unit could control the lighting conditions to a standard deviation of 0.4 lx, providing a 280-fold increase in stability (compared with conditions in which the unit was not used). Additionally, it shielded the plant from wind, thus reducing the distortions in images caused by leaf shaking. The distances between consecutive images were calculated, and the use of the sensor unit reduced the RMSE between images to 0.5% (a 20-fold decrease compared with the conditions in which the unit was not used).

The contributions of this study include the following:

1. A new mobile laboratory robot system is proposed, enabling precise measurements using indoor sensors outdoors by positioning the sensor unit to center the crop and shielding the crop from sunlight and wind. In addition, various functional modules can be developed as a result of making the sensor unit independent of the robot;
2. A practical contribution of this study was the development of a mobile laboratory robot prototype and its experimental evaluation in an actual agricultural field to confirm its measurement and driving capabilities and to show the actual illuminance data in the sensor unit.

This system advances agricultural robotics research by incorporating more precise and varied measurements including hyperspectral cameras and contact sensing with a robotic arm.

**Author Contributions:** Conceptualization, S.N. and Y.M.; formal analysis, S.N. and Y.N.; investigation, S.N., Y.M. Y.N. and M.K.; writing—original draft preparation, S.N.; writing—review and editing, Y.N. and M.K.; supervision, W.I. and J.N.; funding acquisition, W.I. and J.N. All authors have read and agreed to the published version of the manuscript.

**Funding:** This study was funded by the Public–Private R&D Investment Strategic Expansion Program (PRISM) of the Cabinet Office, Government of Japan.

**Data Availability Statement:** Restrictions apply to the availability of the data used, which were under license for the current study and hence are not publicly available. Data are, however, available from the corresponding author (S.N.) upon reasonable request and with permission.

**Conflicts of Interest:** The authors have no conflict of interest to declare.

## Appendix A

All sensor data and some figures obtained in the experiment are shown below.

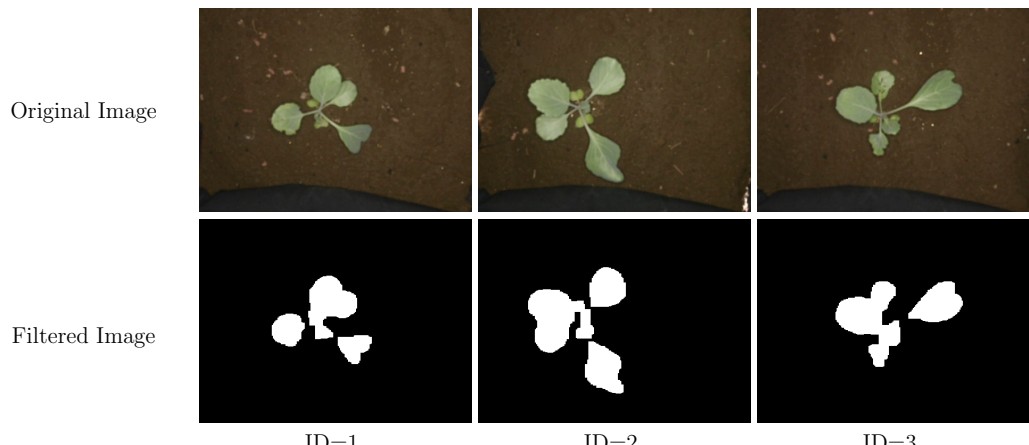

**Figure A1.** Partial results of cropping the cabbage area to white using a LAB color filter.

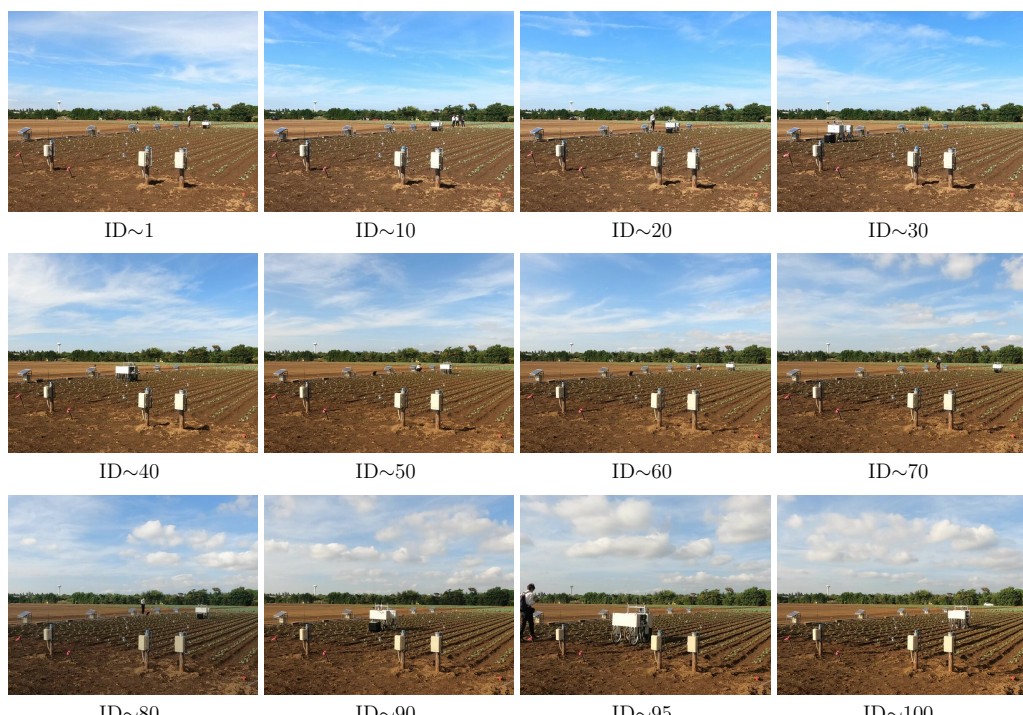

**Figure A2.** Time-lapse photographs of the robot's movement and measurement experiment.

**Table A1.** The soil electric conductivity ($G$), soil permittivity ($\epsilon$), soil temperature ($t$), projected leaf area of the crop ($S$), and RTK-GNSS coordinate at each measurement point from ID = 1 to ID = 50.

| ID | $G$ [dS/m] | $\epsilon$ | $t$ [C] | $S$ [cm$^2$] | Longitude [deg] | Latitude [deg] |
|----|-----------|------------|---------|--------------|-----------------|----------------|
| 1 | 0.15 | 27.1 | 15.1 | 93.7 | 140.104734325 | 36.024414855 |
| 2 | 0.16 | 26.8 | 14.8 | 147.7 | 140.104731293 | 36.024412277 |
| 3 | 0.16 | 26.7 | 15.4 | 116.0 | 140.104728369 | 36.024409555 |
| 4 | 0.14 | 26.8 | 14.3 | 82.7 | 140.104725461 | 36.024406799 |
| 5 | 0.10 | 26.9 | 14.7 | 95.2 | 140.104722848 | 36.024404352 |
| 6 | 0.11 | 27.2 | 12.9 | 78.6 | 140.104702720 | 36.024385612 |
| 7 | 0.07 | 27.6 | 12.8 | 100.3 | 140.104700021 | 36.024382564 |
| 8 | 0.08 | 27.8 | 12.9 | 110.6 | 140.104697467 | 36.024379816 |
| 9 | 0.15 | 27.9 | 14.6 | 80.7 | 140.104694524 | 36.024376763 |
| 10 | 0.09 | 28.1 | 14.4 | 78.8 | 140.104691739 | 36.024374484 |
| 11 | 0.14 | 28.5 | 13.1 | 103.3 | 140.104672059 | 36.024355419 |
| 12 | 0.19 | 28.5 | 14.8 | 76.6 | 140.104669189 | 36.024352688 |
| 13 | 0.23 | 28.4 | 14.5 | 56.8 | 140.104666546 | 36.024350005 |
| 14 | 0.14 | 28.4 | 14.7 | 99.5 | 140.104663762 | 36.024347217 |
| 15 | 0.09 | 28.4 | 14.1 | 93.0 | 140.104660401 | 36.024344652 |
| 16 | 0.14 | 28.5 | 14.7 | 71.6 | 140.104640150 | 36.024325563 |
| 17 | 0.11 | 28.7 | 14.7 | 75.3 | 140.104637733 | 36.024323078 |
| 18 | 0.17 | 28.5 | 14.5 | 50.1 | 140.104634257 | 36.024320177 |
| 19 | 0.15 | 28.5 | 14.2 | 91.8 | 140.104631489 | 36.024317682 |
| 20 | 0.11 | 28.4 | 13.6 | 64.2 | 140.104628858 | 36.024314834 |
| 21 | 0.16 | 28.4 | 12.9 | 83.9 | 140.104603480 | 36.024290790 |
| 22 | 0.13 | 28.5 | 12.6 | 99.5 | 140.104600663 | 36.024288180 |
| 23 | 0.11 | 28.5 | 11.6 | 52.3 | 140.104597404 | 36.024285545 |
| 24 | 0.11 | 28.5 | 12.6 | 86.1 | 140.104594553 | 36.024282751 |
| 25 | 0.09 | 28.5 | 13.2 | 90.4 | 140.104591634 | 36.024280146 |
| 26 | 0.18 | 28.4 | 12.7 | 109.8 | 140.104568988 | 36.024258279 |
| 27 | 0.21 | 28.4 | 12.3 | 113.3 | 140.104566091 | 36.024255666 |
| 28 | 0.11 | 28.5 | 12.5 | 75.6 | 140.104563412 | 36.024253109 |
| 29 | 0.22 | 28.6 | 15.4 | 107.0 | 140.104560340 | 36.024250346 |
| 30 | 0.13 | 28.8 | 13.7 | 108.7 | 140.104557411 | 36.024247724 |
| 31 | 0.18 | 29.4 | 21.8 | 130.0 | 140.104565399 | 36.024228710 |
| 32 | 0.27 | 30.1 | 23.2 | 76.7 | 140.104568693 | 36.024231433 |
| 33 | 0.12 | 30.0 | 14.1 | 152.5 | 140.104571705 | 36.024233990 |
| 34 | 0.14 | 29.8 | 14.2 | 98.0 | 140.104574355 | 36.024236698 |
| 35 | 0.14 | 29.8 | 13.5 | 108.8 | 140.104577590 | 36.024239440 |
| 36 | 0.19 | 30.1 | 13.2 | 97.6 | 140.104603811 | 36.024265234 |
| 37 | 0.16 | 30.1 | 12.9 | 57.8 | 140.104606915 | 36.024267727 |
| 38 | 0.15 | 30.1 | 12.8 | 97.6 | 140.104609900 | 36.024270492 |
| 39 | 0.19 | 30.2 | 13.5 | 96.9 | 140.104612796 | 36.024273093 |
| 40 | 0.14 | 30.2 | 14.2 | 119.3 | 140.104615423 | 36.024276120 |
| 41 | 0.12 | 30.3 | 12.8 | 91.0 | 140.104639361 | 36.024298754 |
| 42 | 0.07 | 30.5 | 11.5 | 58.9 | 140.104642516 | 36.024301224 |
| 43 | 0.08 | 30.5 | 12.4 | 71.1 | 140.104645809 | 36.024303924 |
| 44 | 0.13 | 30.6 | 12.3 | 97.1 | 140.104648653 | 36.024306574 |
| 45 | 0.10 | 30.6 | 12.0 | 117.1 | 140.104651320 | 36.024309309 |
| 46 | 0.06 | 30.5 | 10.7 | 98.8 | 140.104668491 | 36.024325953 |
| 47 | 0.10 | 30.6 | 10.7 | 72.7 | 140.104671250 | 36.024328765 |
| 48 | 0.10 | 30.6 | 11.1 | 94.4 | 140.104674327 | 36.024331907 |
| 49 | 0.07 | 30.7 | 10.6 | 100.1 | 140.104677126 | 36.024334440 |
| 50 | 0.11 | 30.8 | 10.9 | 88.6 | 140.104680218 | 36.024337274 |

**Table A2.** The soil electric conductivity (*G*), soil permittivity (*ε*), soil temperature (*t*), projected leaf area of the crop (*S*), and RTK-GNSS coordinate at each measurement point from ID = 51 to ID = 100.

| ID | *G* [dS/m] | *ε* | *t* [C] | *S* [cm$^2$] | Longitude [deg] | Latitude [deg] |
|----|-----------|------|--------|-------------|-----------------|----------------|
| 51 | 0.08 | 30.8 | 10.9 | 107.9 | 140.104700871 | 36.024356487 |
| 52 | 0.13 | 31.0 | 11.2 | 82.5 | 140.104703772 | 36.024359124 |
| 53 | 0.12 | 31.1 | 11.4 | 55.3 | 140.104706762 | 36.024361824 |
| 54 | 0.12 | 30.8 | 12.0 | 105.5 | 140.104709213 | 36.024364735 |
| 55 | 0.11 | 30.9 | 12.6 | 84.1 | 140.104712116 | 36.024367494 |
| 56 | 0.20 | 30.7 | 14.4 | 113.0 | 140.104732451 | 36.024386851 |
| 57 | 0.25 | 30.7 | 15.0 | 109.7 | 140.104735285 | 36.024389511 |
| 58 | 0.18 | 30.5 | 15.7 | 119.8 | 140.104738254 | 36.024392237 |
| 59 | 0.18 | 30.6 | 15.5 | 127.6 | 140.104741433 | 36.024394851 |
| 60 | 0.16 | 30.5 | 14.3 | 107.9 | 140.104743987 | 36.024397835 |
| 61 | 0.20 | 30.0 | 13.1 | 129.8 | 140.104761636 | 36.024390448 |
| 62 | 0.16 | 30.0 | 13.5 | 104.3 | 140.104758625 | 36.024387540 |
| 63 | 0.17 | 29.9 | 12.9 | 86.4 | 140.104755785 | 36.024384765 |
| 64 | 0.27 | 30.0 | 13.5 | 101.9 | 140.104752897 | 36.024381839 |
| 65 | 0.26 | 29.8 | 14.2 | 80.5 | 140.104749804 | 36.024379347 |
| 66 | 0.24 | 29.8 | 12.7 | 150.6 | 140.104732625 | 36.024362537 |
| 67 | 0.21 | 29.8 | 13.0 | 109.8 | 140.104729762 | 36.024360247 |
| 68 | 0.18 | 29.9 | 13.2 | 101.4 | 140.104726721 | 36.024357490 |
| 69 | 0.12 | 29.8 | 11.9 | 57.2 | 140.104723627 | 36.024354420 |
| 70 | 0.19 | 29.6 | 11.4 | 139.0 | 140.104720921 | 36.024351895 |
| 71 | 0.11 | 29.6 | 11.1 | 107.5 | 140.104700503 | 36.024332580 |
| 72 | 0.22 | 29.6 | 11.8 | 101.4 | 140.104697866 | 36.024329945 |
| 73 | 0.12 | 29.7 | 10.7 | 119.9 | 140.104694816 | 36.024327009 |
| 74 | 0.17 | 29.7 | 11.5 | 73.7 | 140.104692015 | 36.024324435 |
| 75 | 0.14 | 29.8 | 12.1 | 118.7 | 140.104688898 | 36.024321587 |
| 76 | 0.16 | 29.8 | 13.8 | 114.5 | 140.104671421 | 36.024304958 |
| 77 | 0.13 | 29.8 | 13.7 | 119.3 | 140.104668507 | 36.024302184 |
| 78 | 0.16 | 29.6 | 12.5 | 81.4 | 140.104666066 | 36.024299281 |
| 79 | 0.21 | 29.6 | 12.8 | 93.3 | 140.104662877 | 36.024296611 |
| 80 | 0.24 | 29.6 | 13.0 | 109.0 | 140.104659868 | 36.024293971 |
| 81 | 0.10 | 29.4 | 12.0 | 95.6 | 140.104634144 | 36.024269263 |
| 82 | 0.16 | 29.5 | 13.4 | 106.2 | 140.104630643 | 36.024266700 |
| 83 | 0.08 | 29.4 | 12.3 | 113.8 | 140.104627785 | 36.024263854 |
| 84 | 0.12 | 29.4 | 12.3 | 116.5 | 140.104624906 | 36.024261100 |
| 85 | 0.14 | 29.4 | 12.0 | 75.8 | 140.104621914 | 36.024258396 |
| 86 | 0.30 | 29.1 | 12.3 | 88.5 | 140.104598773 | 36.024236690 |
| 87 | 0.19 | 29.2 | 10.3 | 97.9 | 140.104595904 | 36.024234004 |
| 88 | 0.18 | 29.1 | 12.6 | 62.5 | 140.104592747 | 36.024230997 |
| 89 | 0.11 | 29.1 | 11.4 | 86.7 | 140.104589982 | 36.024228736 |
| 90 | 0.11 | 29.2 | 12.3 | 115.0 | 140.104586964 | 36.024225908 |
| 91 | 0.07 | 29.3 | 11.9 | 71.9 | 140.104595227 | 36.024207691 |
| 92 | 0.08 | 29.5 | 11.0 | 72.3 | 140.104598414 | 36.024210574 |
| 93 | 0.05 | 29.6 | 11.8 | 84.2 | 140.104601052 | 36.024213405 |
| 94 | 0.06 | 29.6 | 11.8 | 58.2 | 140.104604099 | 36.024216012 |
| 95 | 0.23 | 29.6 | 13.0 | 53.0 | 140.104607244 | 36.024218799 |
| 96 | 0.03 | 29.8 | 10.9 | 60.7 | 140.104633390 | 36.024242996 |
| 97 | 0.03 | 29.9 | 11.3 | 33.8 | 140.104636232 | 36.024245944 |
| 98 | 0.02 | 30.0 | 11.3 | 12.7 | 140.104639059 | 36.024248591 |
| 99 | 0.03 | 30.0 | 11.9 | 22.3 | 140.104641806 | 36.024251296 |
| 100 | 0.03 | 30.0 | 11.7 | 21.4 | 140.104644980 | 36.024254049 |

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
