# Peer review of "A Mobile Laboratory Robot for Various and Precise Measurements of Crops and Soil in Agricultural Fields: Development and Pilot Study"

_agriculture, doi:10.3390/agriculture13071419_

Round 1

Reviewer 1 Report

The manuscript proposes a "A mobile laboratory robot for various and precise measurements of crops and soil in agricultural fields: development and pilot study". The proposed robot is promising because it has greater data acquisition capability. I appreciate the opportunity to review this manuscript, and I hope my comments can contribute to the manuscript because this is a critical and exciting topic. Please, see my comments in detail below. Please, specific comments are provided below.

Abstrast

1. Clearly articulate the research objective or problem addressed in the study.

2. Emphasize the significance or potential impact of the proposed solution.

Introduction

3. Highlight the significance of the problem being addressed and the potential impact of the research on the field of field monitoring systems.

4. Provide more details on the specific challenges or limitations that the research aims to overcome in the aspect of automated data acquisition.

Overall comments

5. It is suggested to add the description of detailed control pattern to illustrate the superiority of the proposed robot.

6. The article could benefit from a comparison of the proposed solution with existing robots in the field.

Conclusion

7. The contribution of this study should be addressed in the end of Introduction to better attract the attention of peers.

Author Response

We appreciate the reviewer's interest in our research topic.
We are strongly encouraged.

1. (Abstract) Clearly articulate the research objective or problem addressed in the study.

   We have completely rewritten the abstract.
   In the line 1-6, we described the problems and our approaches to solve them.

2. (Abstract) Emphasize the significance or potential impact of the proposed solution.

   In the line 10-14, we summarized the important result and the potential impact.

3. (Introduction) Highlight the significance of the problem being addressed and the potential impact of the research on the field of field monitoring systems.

   In the line 52-56, we added the potential impact and the significant objective.

4. (Introduction) Provide more details on the specific challenges or limitations that the research aims to overcome in the aspect of automated data acquisition.

   In the line 17-23, we summarized the research background.
   In the line 42-51, we added the difficulties and our approaches to overcome them.

5. It is suggested to add the description of detailed control pattern to illustrate the superiority of the proposed robot.

   We added Section 3.5. to show the sequence of the robot control and the throughputs of them.

6. The article could benefit from a comparison of the proposed solution with existing robots in the field.

   We added Section 1.2 to describe the related works.

7. (Conclusion) The contribution of this study should be addressed in the end of Introduction to better attract the attention of peers.

   In the line 89-97, we added the contributions.

Reviewer 2 Report

(1)  Abstract section: Abstract should be rewritten to make it more informative.  

(2)  Introduction section: The objectives of this research should be clearly stated. Please add them.

(3)  Lines 39-41: Is the EC in line 39 the same with that in line 41? The abbreviations should be explained when they were firstly used. Please revised it.

(4)  Lines 49-51: Passive voice should be used, please revise them. Similar sentences in other lines should also be revised.

(5)  Figures 8, 10, 11: It is necessary to mark the positions of “a” and “b” in the figures or below the figures.

Many grammatical errors were found, e.g., “Different objective requires different sensing.” should be “Different objectives require different sensing.”. Moreover, some sentences are too wordy, e.g., lines72-75, please revise them to make them more concise (<30 words).

Author Response

We appreciate the reviewer's careful comments.
We updated the manuscript owing to the comments, as well as the authors checked again the whole text.
We will request the MDPI's English editing before the last submission.

1. Abstract section: Abstract should be rewritten to make it more informative.

   We have completely rewritten the abstract.
   We described the difficulties and our approaches to solve them.
   Besides, we summarized the important results of the experiment.

2. Introduction section: The objectives of this research should be clearly stated. Please add them.

   In the line 17-23, we added a paragraph to explain the background of the research.
   Besides, in the line 42-56, we described the difficulties, our approaches to solve them and important objective.

3. Lines 39-41: Is the EC in line 39 the same with that in line 41? The abbreviations should be explained when they were firstly used. Please revised it.

   In the line 19-20, we explained the abbreviation.

4. Lines 49-51: Passive voice should be used, please revise them. Similar sentences in other lines should also be revised.

   In the line 118-119, we fixed it.
  Besides, we will request the MDPI's English editing before the last submission.

5. Figures 8, 10, 11: It is necessary to mark the positions of "a" and "b" in the figures or below the figures.

   We fixed them.

* Many grammatical errors were found, e.g., "Different objective requires different sensing." should be "Different objectives require different sensing.". Moreover, some sentences are too wordy, e.g., lines72-75, please revise them to make them more concise (<30 words).

  In the line 116 -119, we fixed them.

Reviewer 3 Report

Dear Authors,

In this paper, you present a mobile robot with a deployable measurement unit assessing crop and soil status while providing wind and sunlight cover to improve the measurement itself. The robot's capabilities are tested on an actual field giving promising results.

The topic is of great interest, the overall quality is high and the paper is well-written. In my opinion, this work can be accepted after minor revisions.

Here, I list my main comments:

1. In your plans, the sensing unit should be easily swapped to adapt to different measurements or scenarios. While in absolute favour of this, don't you think that the robot itself and the deployment mechanism limit this possibility to crops similar to cabbage? What about much taller plants? What about fields where the robot cannot drive over the crops?

2. The size of your robot and its element seems driven by the field you are using to test it. Are those dimensions a standard or a widely used convention? Can the robot be easily adapted to different fields?

3. It is not very clear what procedures are already automated or not. I understand that the robot is manually driven (but there are plans to make it autonomous), but what about the measurement procedure? Is it manually activated?

4. Based on fig.5, is 50 cm/s the maximum speed?

5. Does the 4 hours test end because it was planned to sample 100 plants or due to other reasons (e.g., robot battery)?

6. You mention that each sampling last 1 minute but only the last 10 data (20s) are considered. Is there a reason for the extra time?

7. Your test lasted 4 hours and about 1 hour and a half was needed to collect data. What about the remaining time? Is it just for navigating the field? Do you have other details about the timing? For example, what is the average time to sample one plant starting from the previous (so considering lifting the cover, moving, deploying the cover and then measuring)? Also, considering that your test was mostly manually controlled, do you have some indication of if complete automation could improve the system speed?

Minor comments:

- Figures should be placed after they have been mentioned in the text.

- Figure captions should be shorter. Put your comments and discussion about the figure in the text

- In fig.11 there are no (a) and (b) as mentioned in the caption

Author Response

We appreciate the reviewer's insightful comments.
We believe that our manuscript becomes more informative thanks to the comments.

1. In your plans, the sensing unit should be easily swapped to adapt to different measurements or scenarios. While in absolute favour of this, don't you think that the robot itself and the deployment mechanism limit this possibility to crops similar to cabbage? What about much taller plants? What about fields where the robot cannot drive over the crops?

   In the line 386-392, we added a paragraph to discuss the target crops.
   We think taller plants will be ok, but crops that cover the ground will be difficult.

2. The size of your robot and its element seems driven by the field you are using to test it. Are those dimensions a standard or a widely used convention? Can the robot be easily adapted to different fields?

   In the line 380-385, we added a paragraph to discuss the field dimensions.
   We think that the dimension used in this paper is a common value in Japanese cabbage field.
   We cited research on robots with variable tread width for more general dimensions.

3. It is not very clear what procedures are already automated or not. I understand that the robot is manually driven (but there are plans to make it autonomous), but what about the measurement procedure? Is it manually activated?

   We added Section 3.5 to describe the procedures.
   After controlling the robot using joystick towards the target crop, the operator active the measure button.
   The robot automatically moves the sensor unit down to the crop, measures, and move the sensor unit up.

4. Based on fig.5, is 50 cm/s the maximum speed?

   In the line 206-207, we clarified the maximum speed.
   The speed is 50 cm/s limited by software (hardware limit is not strictly tested).
   We confirmed that the PI controller could track up to 50 cm/s.

5. Does the 4 hours test end because it was planned to sample 100 plants or due to other reasons (e.g., robot battery)?

   The reason of the 4 hours is that it was planned to sample 100 plants.
   In the line 321-323, we discussed the battery capability.

6. You mention that each sampling last 1 minute but only the last 10 data (20s) are considered. Is there a reason for the extra time?

   In the line 230-232, we explained the reason why we used last 10 data.
   We wait for the sensor value to become stable as the sensor body adjusts to the ground temperature.

7. Your test lasted 4 hours and about 1 hour and a half was needed to collect data. What about the remaining time? Is it just for navigating the field? Do you have other details about the timing? For example, what is the average time to sample one plant starting from the previous (so considering lifting the cover, moving, deploying the cover and then measuring)? Also, considering that your test was mostly manually controlled, do you have some indication of if complete automation could improve the system speed?

   We added Section 3.5 to describe the throughputs of the system.
   In the line 363-370, we discussed the improvement of the speed by the automated driving.

* Figures should be placed after they have been mentioned in the text.

  We fixed them.

* Figure captions should be shorter. Put your comments and discussion about the figure in the text

  We fixed them.

* In fig.11 there are no (a) and (b) as mentioned in the caption

  We fixed them.

Round 2

Reviewer 2 Report

The comments and suggestions have been addressed by authors and the manuscript has been significantly improved.  Currently it can be considered to be accepted after a moderate editing of English language.

Moderate editing of English language is required.

Author Response

We appreciated the reviewer's careful checking again.

The comments and suggestions have been addressed by authors and the manuscript has been significantly improved.  Currently it can be considered to be accepted after a moderate editing of English language.

We completed the MDPI's English editing.
The updates are highlighted in red.
We checked that our intended meaning was retained.